# *FTO* Regulated Intramuscular Fat by Targeting *APMAP* Gene via an m^6^A-YTHDF2-dependent Manner in Rex Rabbits

**DOI:** 10.3390/cells12030369

**Published:** 2023-01-19

**Authors:** Gang Luo, Tingting Hong, Lin Yu, Zhanjun Ren

**Affiliations:** College of Animal Science and Technology, Northwest A & F University, Xianyang 712100, China

**Keywords:** Rex rabbits, intramuscular fat, m^6^A methylation, *FTO* gene, *APMAP* gene

## Abstract

N6-methyladenosine (m^6^A) regulates fat development in many ways. Low intramuscular fat (IMF) in rabbit meat seriously affects consumption. In order to improve meat quality, we explored the law of IMF deposition. *FTO* could increase the expression of *APMAP* and adipocyte differentiation through methylation. However, interference *YTHDF2* can partially recover the influence of interference *FTO* on the *APMAP* gene and adipocyte differentiation. *APMAP* promoted the differentiation of adipocytes. Analysis of IMF and *APMAP* expression showed IMF content is positive with the expression level of the *APMAP* gene (*p* < 0.01). Conclusion: Together, *FTO* can regulate intramuscular fat by targeting the *APMAP* gene via an m^6^A-YTHDF2-dependent manner in Rex rabbits. The result provides a theoretical basis for the molecular breeding of rabbits.

## 1. Introduction

Meat contains most minerals, vitamins, essential fatty acids, and other nutrients that human beings need [1]. The flavor of meat food is the main factor affecting consumers’ choice and fatty acids in intramuscular fat can affect the flavor of meat food [2,3]. The content and composition of IMF is a very important symbol of meat quality and IMF influences the flavor, juiciness, and tenderness of meat [4]. Adipocyte differentiation and proliferation form fat deposition. Therefore, understanding the mechanism of adipogenesis could provide a scientific basis for breeding rabbits with high intramuscular fat.

M^6^A was discovered in the 1970s [5,6] and involved almost all aspects of RNA metabolism. Demethylases (FTO) [7,8], methylase [5], and methylation recognition enzyme (YTHDF2) [9,10,11,12] are the main factors causing m^6^A modification. The *FTO* gene is the first gene [13,14] that regulates fat deposition [15,16]. Studies indicated knockout of *FTO* inhibited fat deposition in mouse liver [17] and overexpression of *FTO* promoted adipogenesis in cells [18]. *YTHDF2* has been found to regulate the translation of m^6^A containing mRNA [19]. However, it is not clear whether *FTO* and *YTHDF2* regulate adipogenesis through m^6^A modification. So, it is important to explore the regulatory pathway of *FTO* on fat deposition.

*APMAP* contains 415 amino acids [20]. Studies found *APMAP* participated in the material exchange between the environment and mature adipocytes and regulates adipogenesis [21]. In addition, interference *APMAP* reduced fat deposition [21,22], which indicated *APMAP* was a crucial factor in adipogenesis. However, the role of *APMAP* in adipogenesis has not been reported yet.

In this study, we found that the expression level of the *APMAP* gene was higher in the muscle tissue of rabbits with high fat content. Further study showed *FTO* knockdown decreased *APMAP* expression by identification function of *YTHDF2*. In addition, we found that *APMAP* promoted adipocyte differentiation. Finally, the results of tissue PCR and intramuscular fat content showed that they were positively correlated (*r* = 0.844, *p* < 0.01). Our study provided a new way to breed Rex rabbits with high-quality meat.

## 2. Material and Methods

### 2.1. Animals

The adipose tissues were isolated from newborn rabbits’ perirenal fat after rabbits were slaughtered humanely. Adipose tissues were cut in PBS and digested in a 37 ℃ water bath for 1 h. Finally, preadipocytes were obtained after filtering and centrifuging the mixture. Longissimus lumborum and perirenal fat were rapidly separated from 18 female Rex rabbits who were aged 35 days, 75 days, and 165 days. After separating the sample, we quickly put it in liquid nitrogen for 15 min and kept it in the −80 ℃ refrigerator for a long time to extract mRNA. In addition, another 15 g longissimus lumborum was isolated to keep in the −20 ℃ refrigerator. These rabbits were raised under standard conditions (Farm of Northwest Agriculture and Forestry University Yangling, Shaanxi, China).

### 2.2. Ethical Statement

All research involving animals was conducted following the Regulations for the Administration of Affairs Concerning Experimental Animals and approved by the Institutional Animal Care and Use Committee in the College of Animal Science and Technology, Northwest A&F University, Yangling, China, under permit No. DK-2019008.

### 2.3. Cell Experiment

The method of cell culture is consistent with that in the previous literature [23]. In short, we digested the tissue into a single cell with 0.25% collagenase type I and filtered the cells with a 40 µm cell sieve. Then, the cells were cultured at 37 ℃ for a period of time. Finally, the induced differentiation solution (DM/F12, 0.5 mM 3-isobutyl-1-methylxanthine, 1.7 μM insulin, 10% fetal bovine serum, 1 μM dexamethasone, and 2% penicillin-streptomycin) and maintained differentiation solution (DM/F12, 1.7 mM insulin, 10% fetal bovine serum, and 2% penicillin-streptomycin) were used to differentiate the cells in the incubator (37 ℃, 5% CO_2_). We used Lipofectamine 2000 (Invitrogen, Carlsbad, CA, USA) to transfect the synthesis into cells.

### 2.4. Oil Red O Staining and Measurement of Triglyceride Content

Oil red O staining solution (Solarbio, Beijing, China) was used to stain fat droplets of adipocytes. In short, we fixed them with 4% formaldehyde for 30 min after PBS washing cells. Cells were stained in a dark environment with oil red O solution for 30 min after discarding the formaldehyde solution. After dyeing, we washed cells with water and took pictures under a microscope. Finally, 200 uL isopropanol was added and the OD value was measured at 510 nm wavelength of the microplate reader. A TG Assay Kit (Applygen, Beijing, China) was used to obtain the Intracellular triglyceride (TG) content. Firstly, the standard was prepared and diluted to obtain standards of different concentrations. Secondly, the cells were lysed using cell lysate, and the mixture was heated at 70 ℃ for 10 min before centrifugation. Then, OD values of the supernatant and the standard samples of various concentrations were measured using the microplate reader. Finally, we calculate the content of triglyceride according to the concentration of the standard through the standard curve.

### 2.5. RT-qPCR

Total RNA was obtained by TRIzol Reagent (Invitrogen, Carlsbad, CA, USA). A NanoDrop 2000 spectrophotometer (Thermo, Waltham, MA, USA) was used to assess RNA quality. RT-qPCR was performed by the previous method [24] and the primers are in Table 1.

### 2.6. Gene-specific m^6^A qPCR

The methylation level was determined with a Magna MeRIP m^6^A Kit (Millipore, Darmstadt, Germany). Briefly, fragmented RNA binds to the antibody at 4 degrees for 12 h. After using the reaction mixture of A/G magnetic beads, we used the RNeasy kit (Qiagen, Shanghai, China) to recover the methylated RNA. Finally, the methylation level of fragmented RNA and enriched RNA was detected by qPCR.

### 2.7. Western Blotting

Total protein was extracted with the RIPA Lysis Buffer (CWBIO, Jiangsu, China) and Protease Inhibitor Cocktail (CWBIO, Jiangsu, China). BCA Protein Assay Kit was used to quantify the protein content. Western blotting drew on the previous literature methods [25]. The protein solution was heated at 100 ℃ for 10 min after adding the loading buffer. Then, 40 mg proteins were added in 4–12% SDS-polyacrylamide gels for electrophoresis (1.5 h). After electrophoresis, we used a semi-dry membrane rotator to rotate the membrane for 25 min at 18 volts. The membrane was incubated for 12 h in primary antibody solution at 4 ℃ and was incubated for 1 h in secondary antibody solution at room temperature. Finally, protein bands were obtained with a Bio-Rad GelDoc system equipped with a Universal Hood III (Bio-Rad), and a gray value was used to quantify the protein expression level.

### 2.8. Measurement of IMF

IMF was measured using Soxhlet petroleum-ether extraction [26]. In brief, the meat sample was chopped into minced meat, then dehydrated in an oven to a constant amount, cooled and crushed. We accurately weighed about 1.0 g of the treated sample (F) into a filter paper cylinder and then dehydrated it to a constant amount in an oven at 105 ℃ so that the total mass was F1. Subsequently, the sample was treated in a Soxhlet extractor and was dried in an oven at 105 ℃ to a constant amount (F2). IMF = (F1 − F2)/F × 100%.

### 2.9. Statistical Analysis

GraphPad Prism software 5.0 (GraphPad, Santiago, CA, USA) was used to assess all data. One-way analysis of variance (ANOVA) and two-tail Student’s *t*-test were used in this study. A general linear model was fitted in the R software environment [27] to develop prediction models for *APMAP* expression level and IMF. The model is as follows, Intramuscular fat = *APMAP* expression level + Age. Marginal and conditional *R^2^* values were calculated for the model. *p* < 0.05 and *p* < 0.01 were significant and highly significant, respectively.

## 3. Results

### 3.1. Methylation Modification of APMAP and APMAP Expression in Rabbits during Growth Periods

Based on previous MeRIP-seq results (the raw data are published and the results did not appear in published articles [28]), we found that mRNA of *APMAP* was methylated in both muscle and fat tissue (Figure 1A). As shown in Figure 1A, the methylation of *the APMAP* gene in perirenal fat and muscles occurred at multiple loci and there were significant differences. The expression level of *APMAP* in fat and muscle increased with age (Figure 1B,C).

### 3.2. Preadipocytes Deletion of FTO Inhibits Adipocyte Differentiation

To investigate the role of *FTO* during adipogenesis, the *FTO* gene was interfered. The results of qPCR and WB showed that the experiment was successful (Figure 2A–C). Silencing of *FTO* significantly inhibited adipogenic differentiation (Figure 2D,E). The content of Triglyceride also indicated that knockout of *FTO* significantly inhibited adipogenic differentiation (Figure 2F). In addition, PPARγ C/EBPα, and FABP4 expression were inhibited when *FTO* was interfered (Figure 2G–I).

### 3.3. Upregulation of FTO Increased Preadipocyte Differentiation

Tofurther make clear the role of *FTO*, the function studies were conducted by using pcDNA3.1 + FTO. As shown in Figure 3A–C, *FTO* was successfully upregulated. When *FTO* was overexpressed, the increased lipid droplets were observed (Figure 3D,E) and the increased content of Triglyceride was measured (Figure 3F). In addition, adipogenic marker genes were significantly promoted (Figure 3G–I).

### 3.4. The Deletion of YTHDF2 Partially Restored Adipocyte Differentiation of FTO Depleted Cells

To explore the molecular mechanisms of *YTHDF2* on *FTO* regulating adipocyte differentiation, *YTHDF2* was inhibited in *FTO* siRNA adipocytes. Western blot assay and qPCR were shown and our operation was effective (Figure 4A–D). The content of triglyceride and lipid droplets was partially restored in *FTO* siRNA adipocytes (Figure 4E–G). As expected, adipogenic marker gene expression levels were significantly higher than that in *FTO* siRNA adipocytes when *YTHDF2* was knocked out (Figure 4H–J).

### 3.5. FTO and YTHDF2 Interact to Regulate APMAP Expression

As shown in Figure 5A–C *APAMP* expression was significantly lower after interfering with the *FTO* gene (*p* < 0.01) whereas fold enrichment of the *APMAP* gene was significantly higher (*p* < 0.01) (Figure 5E). As expected, overexpression of *FTO* achieved the opposite result (Figure 5D). After knocking out the *YTHDF2* gene in *FTO* siRNA adipocytes. *APAMP* expression was significantly higher (*p* < 0.01) (Figure 5F–H) whereas fold enrichment of methylation modification was significantly lower (*p* < 0.01) (Figure 5I) than that in *FTO* siRNA adipocytes.

### 3.6. APMAP Is Essential for Adipogenesis In Vitro

We knocked out the *APMAP* gene in preadipocytes and found knockout was effective (Figure 6A–C). Contents of triglycerides and lipid droplets indicated that *APMAP* could promote the differentiation of adipocytes (Figure 6D–F). As expected, the same conclusion is drawn from the verification results of adipogenic marker genes (Figure 6G–I).

### 3.7. Correlation Analysis between Intramuscular Fat Content and APMAP mRNA Expression

The correlation between IMF and *APMAP* expression was presented in Table 2. For the model, intramuscular fat content and expression level of the *APMAP* gene (*p* < 0.01) were significant. The marginal *R^2^* = 0.9461 and a conditional *R^2^* = 0.9345 (*p* < 0.01).

## 4. Discussion

In recent years, scientists have been very keen on the exploration and research of fat deposition mechanisms. However, mRNA m^6^A regulates fat development poorly. In our study, we found that *FTO* regulated adipocyte differentiation through interference and overexpression of *FTO*. Previous studies have shown that *FTO* regulated fat deposition by promoting adipocyte differentiation [29,30]. In addition, *FTO* promoted the activity of *FAS* and *PPARγ* genes [31]. A previous study found that restricting feed significantly decreased *FTO* mRNA levels and insulin levels [32,33]. In addition, the level of *IGF-1* decreased in *FTO*-specific deficient mice [34]. These results indicated *FTO* can promote insulin secretion. Studies found insulin can stimulate the phosphorylation of *AKT* and *IRS-1* and regulate the function of *AKT* and *IRS-1*. [35,36]. Interference *APMAP* significantly reduced the expression of *P-AKT* and *pIRS-1* [37]. In summary, *FTO* may regulate the expression level of the *APMAP* gene. In our study, *APMAP* expression decreased significantly whereas the methylation level of the *APMAP* gene increased significantly after transfection *si-FTO* (Figure 5). These results indicated that *FTO* affected the expression level of the *APMAP* gene by m^6^A methylation.

The modification of m^6^A to mRNA transcripts should be recognized by specific proteins, which are m^6^A readers [38]. *YTHDF3*, *YTHDF1,* and *YTHDF2* were identified as m^6^A readers [10,39]. *YTHDF1* and *YTHDF3* played an important role in protein synthesis and mRNA translation [40,41]. *YTHDF2* participated in recognizing and destabilizing m^6^A-containing mRNA [10]. A previous study revealed that *FTO* regulated adipogenesis via m6A-YTHDF2 dependent mechanism [25]. In addition, The study also demonstrated epigallocatechin gallate targets *FTO* and *FTO* inhibited adipogenesis by *YTHDF2* [42]. These results revealed that *YTHDF2* can recognize *FTO* and regulate the expression of *FTO*. In our study, the data of MeRIP-seq also showed that the *APMAP* gene is regulated at multiple sites by m^6^A methylase (Figure 1A). Interference of *YTHDF2* partially rescued the expression level of *APMAP* in *FTO*-depleted cells (Figure 4). The methylation level of *APMAP* gene was significantly recovered (Figure 5I). These results indicated *FTO* regulated *APMAP* gene expression by *YTHDF2* gene.

*APMAP* is a regulatory factor related to adipocyte differentiation [43]. Interference *APMAP* inhibited adipocyte differentiation [22]. *APMAP* caused insulin resistance through IRS-1/insulin sensitive genes/free fatty acids pathway [44,45,46]. The activity of *FAS* can be improved by insulin in adipocytes [47,48] and high *FAS* expression significantly increased diacylglycerol deposition and caused obesity [49]. At the same time, *FAS* expression was positively correlated with body fat levels in many mammals [50]. In this study, we also found that *APMAP* promoted adipocyte differentiation (Figure 6).

The study showed interfering with *APMAP* inhibited the differentiation of preadipocytes and lipid droplet formation [21]. In Table 2, we found intramuscular fat content and *APMAP* expression (*p* < 0.01) were significant. So, a high expression level of the *APMAP* gene means high intramuscular fat content. In addition, we found that the expression level of the *APMAP* gene increases with age. Studies also showed intramuscular fat content of rabbits increased with age [51]. These results indicated *APMAP* can regulate the intramuscular fat content of Rex rabbits. In summary, *FTO* regulated the expression of *APMAP* through *YTHDF2* recognition and *APMAP* can promote intramuscular fat deposition(Figure 7).

## 5. Conclusions

In summary, we found *FTO* promoted the expression level of the *APMAP* gene by *YTHDF2* recognition. *APMAP* promoted differentiation of adipocytes and *APMAP* expression was positively correlated with IMF content. *FTO* promotes intramuscular fat by targeting.

*APMAP* gene via *YTHDF2* recognition (Figure 7). The mechanism of IMF may provide a theoretical basis for the molecular breeding of rabbits with high meat quality.

## Figures and Tables

**Figure 1 cells-12-00369-f001:**
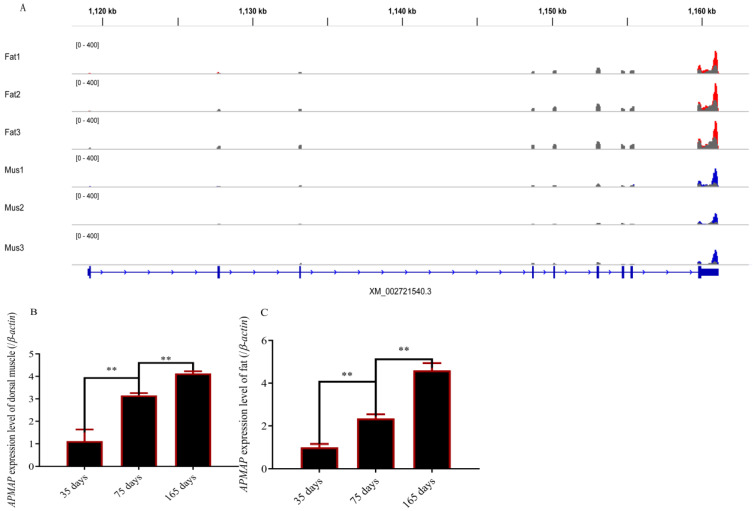
Methylation modification and *APMAP* expression in muscle and fat tissues. (**A**) Examples of dynamic methylated with m^6^A peaks in *APMAP* gene (According to our m^6^A sequencing data and we selected an m^6^A modified gene from them. The article was published in *Biology journal* [28]. This figure does not exist in the published article. This figure only plays the role of leading out the research in this paper.); (**B**) *APMAP* expression in dorsal muscles at 35, 75, and 165 days of age; (**C**) *APMAP* expression in fat tissues at 35, 75, and 165 days of age ( ** *p* ≤ 0.01).

**Figure 2 cells-12-00369-f002:**
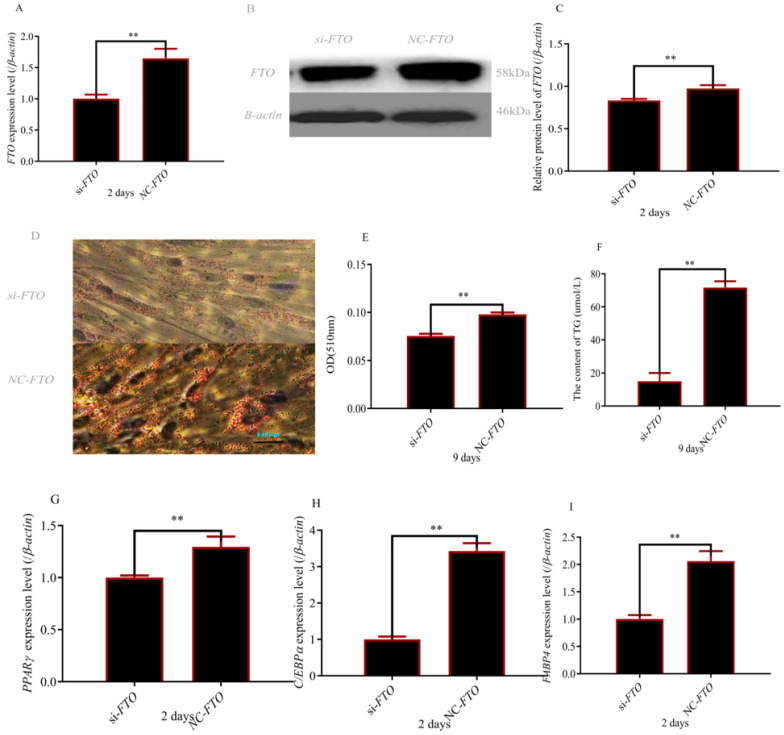
Interference of *FTO* reduced preadipocyte differentiation: (**A**) *FTO* expression level after transfecting with *si-FTO* and NC; (**B**,**C**) FTO protein level after transfecting with *si-FTO* and NC; (**D**) lipid droplets (magnifications = 10 × 10); (**E**) lipid droplet content; (**F**) Triglyceride content; (**G**–**I**) PPARγ, CEBPα and FABP4 expression after transfecting with *si-FTO* and NC ( ** *p* ≤0.01).

**Figure 3 cells-12-00369-f003:**
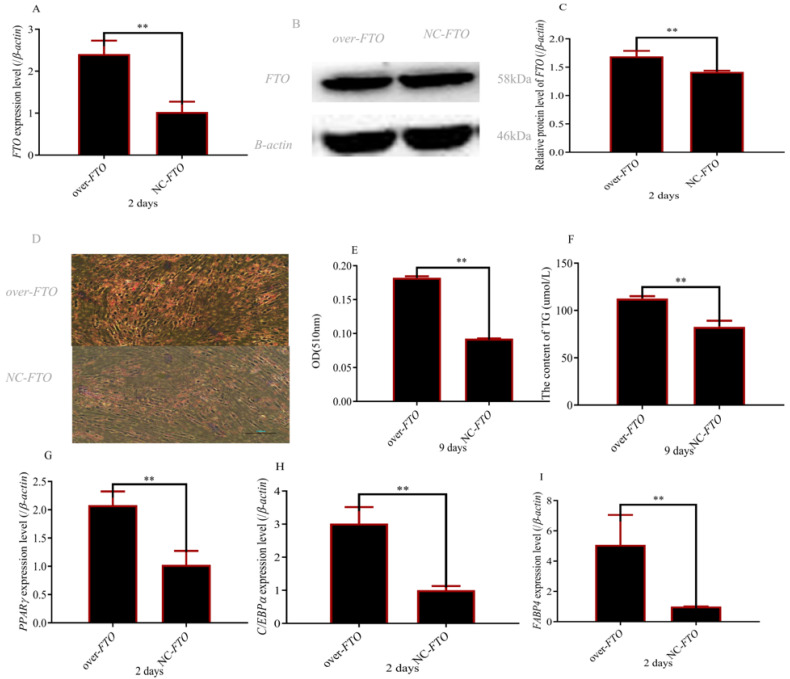
Overexpression of *FTO* gene promoted preadipocyte differentiation: (**A**) *FTO* expression level after overexpression of *FTO*; (**B**,**C**) FTO protein level after overexpression of *FTO*; (**D**,**E**) size and content of lipid droplets (magnifications = 10 × 10); (**F**) Triglyceride content; (**G**–**I**) expression levels of PPARγ, CEBPα and FABP4 after overexpression of *FTO* ( ** *p* ≤0.01).

**Figure 4 cells-12-00369-f004:**
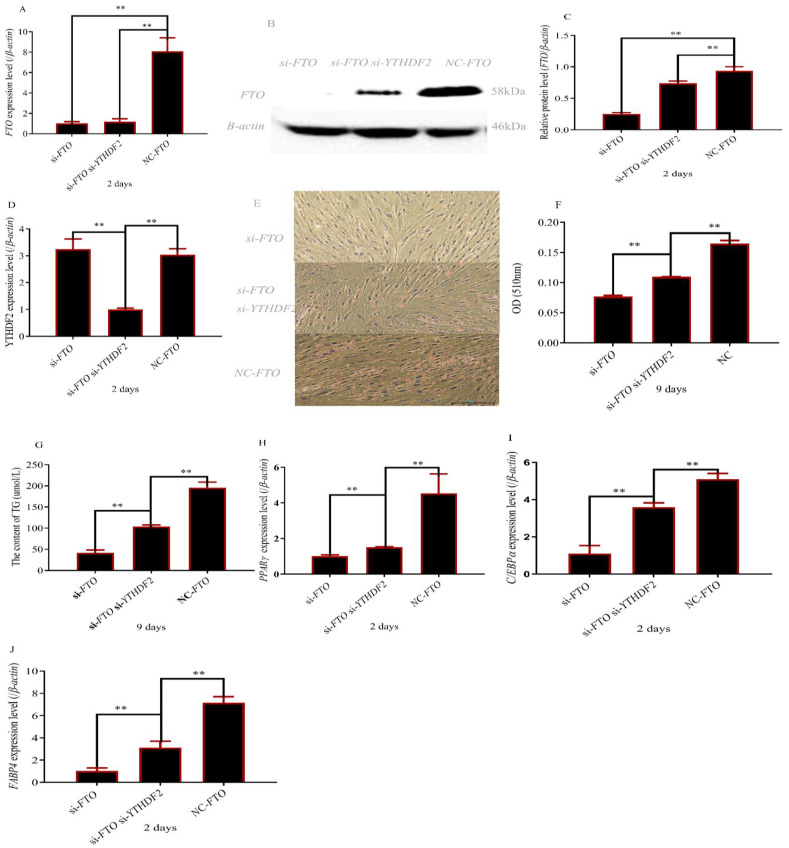
Inhibition of *YTHDF2* reduced the inhibitory effect of interference *FTO* gene on rabbit preadipocyte differentiation. (**A**) *FTO* expression after transfecting with *si-FTO*, *si-YTHDF2* and NC; (**B**,**C**) FTO protein level after transfecting with *si-FTO*, *si-YTHDF2* and NC; (**D**) *YTHDF2* expression after transfecting with *si-FTO*, si-YTHDF2 and NC; (**E**) size of lipid droplets (magnifications = 10 × 10); (**F**) content of lipid droplets; (**G**) Triglyceride content; (**H**–**J**) expression levels of PPARγ, CEBPα and FABP4 after transfecting with *si-FTO*, *si-YTHDF2* and NC; ( ** *p* ≤ 0.01).

**Figure 5 cells-12-00369-f005:**
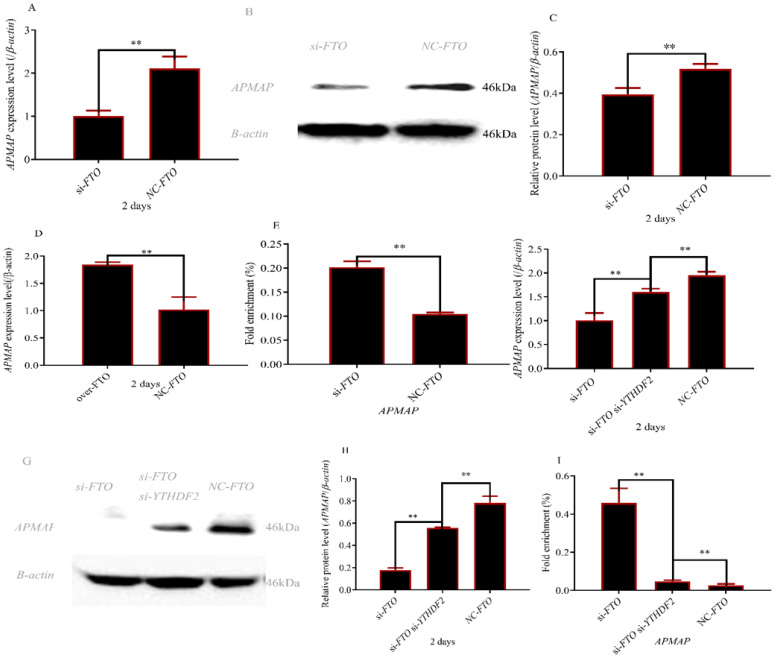
*FTO* upregulated *APMAP* expression level in an m6A-YTHDF2 manner. (**A**) *APMAP* expression level after interfering with *FTO* gene; (**B**,**C**) APMAP protein level after interfering with *FTO* gene; (**D**) *APMAP* expression level after overexpression of *FTO*; (**E**) fold enrichment of *APMAP gene* after interfering with *FTO* gene; (**F**) *APMAP* expression level after interfering with *FTO* and *YTHDF2* genes; (**G**,**H**) APMAP protein levels after interfering with *FTO* and *YTHDF2* genes; (**I**) fold enrichment of *APMAP gene* after interfering with *FTO* and *YTHDF2* genes; ( ** *p* ≤ 0.01).

**Figure 6 cells-12-00369-f006:**
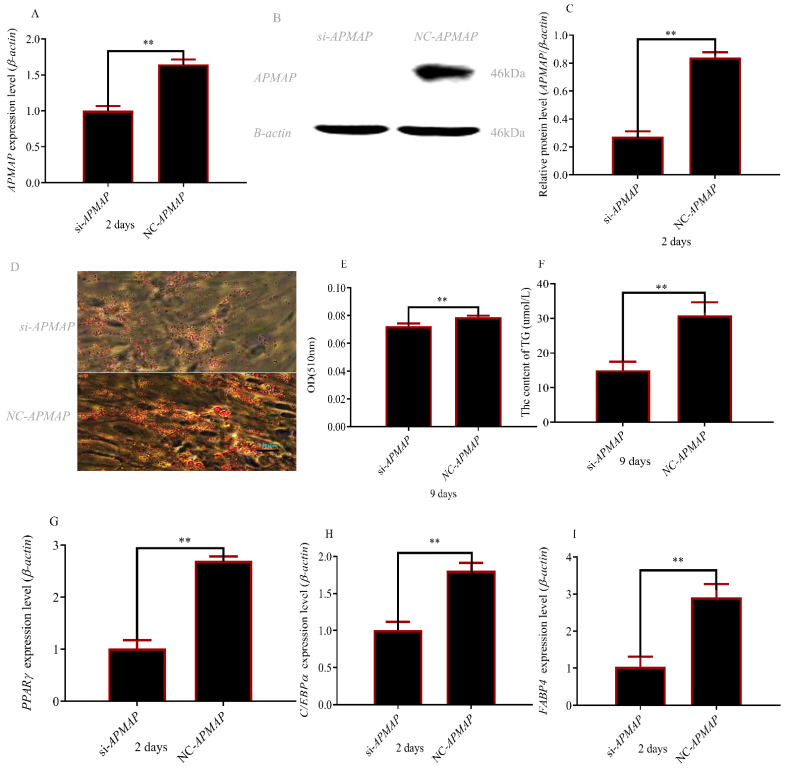
Inhibition of *APMAP* gene inhibited rabbit preadipocyte differentiation: (**A**) *APMAP* expression level after transfecting with *si-APMAP* and NC; (**B**,**C**) APMAP protein level after transfecting with *si-APMAP* and NC; (**D**) size of lipid droplets (magnifications = 10 × 10); (**E**) content of lipid droplets; (**F**) Triglyceride content; (**G–I**) PPARγ, CEBPα, and FABP4 expression after transfecting *si-APMAP* and NC ( ** *p* ≤ 0.01).

**Figure 7 cells-12-00369-f007:**
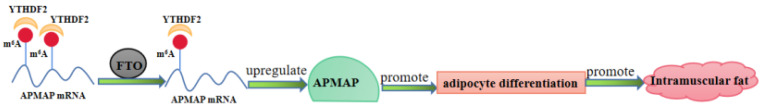
Regulatory mechanisms by which *FTO* affects intramuscular fat of Rex rabbits through m^6^A/YTHDF2/APMAP pathway.

**Table 1 cells-12-00369-t001:** Primers.

Gene Name	Primer Sequence (5′-3′)	(Tm/°C)	CG%	(Product Size/bp)
*FTO*	ATCCCGATCTCTCACCACAC	60	55	183
	ACATCTGCGGACCATACAAA		45	
*YTHDF2*	CAGACACAGCCATTGCCTCCAC	60	59	122
	CCGTTATGACCGAACCCACTGC		59	
*APMAP*	GCTGCTGGATTCTCCCATAG	60	55	163
	AAACATCACGTCCCCGATAT		45	
*PPARγ*	GAGGACATCCAGGACAACC	61	58	168
	GTCCGTCTCCGTCTTCTTT		53	
*β-actin*	GGAGATCGTGCGGGACAT	61.4	61	318
	GTTGAAGGTGGTCTCGTGGAT		52	
*C/EBPα*	GCGGGAACGAACAACAT	64	53	172
	GGCGGTCATTGTCACTGGTC		6	
*FABP4*	GGCCAGGAATTTGATGAAGTC	61.4	48	140
	AGTTTATCGCCCTCCCGTT		53	
*si-YTHDF2*	CAUGAAUACUAUAGACCAATT		29	
	UUGGUCUAUAGUAUUCAUGTT		29	
*si-FTO*	GCAGCUGAAAUAUCCUAAATT		33	
	UUUAGGAUAUUUCAGCUGCTT		33	
*si-APMAP*	GUGGAAAGGCUAUUUGAAATT		33	
	UUUCAAAUAGCCUUUCCACTT		33	
*over-FTO*	GCTAGCGCCACCATGAAGC		63	
	CTCGAGCTAAGGCTTTGCTTCC		55	
*Negative Control*	UUCUCCGAACGUGUCACGUTT		48	
	ACGUGACACGUUCGGAGAATT		48	

**Table 2 cells-12-00369-t002:** *APMAP* expression affected IMF in Rex rabbits (regression coefficients, standard errors, and probability levels).

Model	Coefficient	Std Error	*p*-Value
Intercept	0.0001347	0.0199996	0.99472
*APMAP* Expression in longissimus lumborum	0.0007015	0.0001804	0.00164

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
