# Peer review of "FTO Regulated Intramuscular Fat by Targeting APMAP Gene via an m6A-YTHDF2-dependent Manner in Rex Rabbits"

_cells, 2023, doi:10.3390/cells12030369_

Round 1

Reviewer 1 Report

The authors studied the role of FTO/m6A/YTHDF2/APMAP signaling in intramuscular fat in Rex rabbits.

Comments need to be addressed:

1)  2.1. Animals and tissue collection: Details should be provided on how the different tissues were isolated from the newborn rabbits.

2)  2.3. Cell culture and cell transfection: Details should be provided for how the tissues were digested into a single cell and filtered, induced differentiation solution and maintained differentiation solution were used to differentiate the cells!. What are the recipe for these solutions, conditions for the differentiation?...etc

3)  2.4. Oil Red-O staining and measurement of triglyceride content: Details should be provided; concentration used, time of incubation, etc..

4)  Data in Table 1 (3rd row) should be aligned and I advise to add the %GC

5)  The negative control in Table 1; what gene it represents?

6)  Some of the genes in Table 1 lacks the Tm and product size

7)  The resolution is very poor for Fig 1A, should be enhanced!

8)  Generally, for all bar chart figures, I don’t see that colours illustrate any additional info for the bar charts, the authors don’t need a legend that is also represented on the x-axis. Also, the y-axis should be represented as fold change!

9)  Generally, for figures some labels are either grey in color or not of the same font size, please adjust!

10)  Fig 2A: Why the FTO expression level was normalized over B-actin, is this the NC?!

11)  Fig2B: the blot is overexposed, Please adjust!

12)  All figures of the Oil Red-O staining are too dark!!. Please adjust!

13)  Scale bars and magnifications should be provided for Figs 2D, 3D, 4E, and 6D

14)  Looking to the si-FTO in fig 1b and fig 4b, it seems for 4b, FTO is completely knocked down which is not the case in 1b. How the authors could explain that?!

15)  For fig 5B and 5G, the blot for FTO from the same setup should be included as a control for the knockdown, since there is an inconsistency in the knockdown based on figs  fig 1B and fig 4B.

16)  The blots for actin in fig 4B and 5G might be the same blot, 5G seems like it was flipped and compressed, Please check and clarify!

17)  Please check the reference list; ex; ref 5, 17, 20, 23, and others

18)  The English should be revised by a native English speaker

Author Response

Thank you very much for your guidance to my manuscript. We have made relevant revisions according to the proposed guidance. I hope to be approved by editor and reviewer. The specific reply is as follows: The authors studied the role of FTO/m6A/YTHDF2/APMAP signaling in intramuscular fat in Rex rabbits. Comments need to be addressed: 1)  2.1. Animals and tissue collection: Details should be provided on how the different tissues were isolated from the newborn rabbits. We have corrected it 2)  2.3. Cell culture and cell transfection: Details should be provided for how the tissues were digested into a single cell and filtered, induced differentiation solution and maintained differentiation solution were used to differentiate the cells!. What are the recipe for these solutions, conditions for the differentiation?...etc We have corrected it 3)  2.4. Oil Red-O staining and measurement of triglyceride content: Details should be provided; concentration used, time of incubation, etc.. We have corrected it 4)  Data in Table 1 (3rd row) should be aligned and I advise to add the %GC We have corrected it 5)  The negative control in Table 1; what gene it represents? The universal negative control has no homology with the target gene sequence. So, The controls of FTO, YTHDF2 and APMAP all use negative control. 6)  Some of the genes in Table 1 lacks the Tm and product size The sequence without Tm and product size is the sequence of interfering RNA, which does not involve PCR experiment, so there is no Tm and product size. 7)The resolution is very poor for Fig 1A, should be enhanced! We have corrected it 8)  Generally, for all bar chart figures, I don’t see that colours illustrate any additional info for the bar charts, the authors don’t need a legend that is also represented on the x-axis. Also, the y-axis should be represented as fold change! We have corrected it,y-axis represents the relative expression quantity, not fold change. 9) Generally, for figures some labels are either grey in color or not of the same font size, please adjust! We have corrected it 10)  Fig 2A: Why the FTO expression level was normalized over B-actin, is this the NC?! B-actin is housekeeping gene. The housekeeper gene is used as a control, and its expression is constant. In this way, you can make standardized values by comparing it with your gene. All quantification must be accompanied by the housekeeper gene to eliminate interference. NC is the negative control group of the experiment. 11)  Fig2B: the blot is overexposed, Please adjust! We have corrected it 12)  All figures of the Oil Red-O staining are too dark!!. Please adjust! We have corrected it 13)  Scale bars and magnifications should be provided for Figs 2D, 3D, 4E, and 6D We have corrected it 14)  Looking to the si-FTO in fig 1b and fig 4b, it seems for 4b, FTO is completely knocked down which is not the case in 1b. How the authors could explain that?! In Figure 1B, under the same conditions, only the control group and the experimental group are needed for the experiment, so the 6 well plate is used. In Figure 4B, the same experiment requires three groups, so the 12 well plate is used. The culture vessels of the two groups are different, the number of transfected cells is different, and the transfection efficiency is not exactly the same, so the results are not exactly the same. 15)  For fig 5B and 5G, the blot for FTO from the same setup should be included as a control for the knockdown, since there is an inconsistency in the knockdown based on figs  fig 1B and fig 4B. In Figure 1B and fig 5B, we used 6 well plate to culture cells. But in figure 5G and fig 4B, we used 12 well plate to culture cells. Preadipocytes cannot be passed down all the time, so we can only use them now. These four results are from cells of different individuals, and there are differences between individuals. In addition, The number of cells of Figure 1B/ fig 5B and Figure 4B/ fig 5G during transfection is different, so the transfection efficiency cannot be identical. In summary, the protein bands of these four results cannot be identical. 16)  The blots for actin in fig 4B and 5G might be the same blot, 5G seems like it was flipped and compressed, Please check and clarify! The B-ACTIN of 5G and 4B has the same imprint. B-ACTIN is the housekeeping gene, which can be used not only as the housekeeping gene of APMAP, but also as the housekeeping gene of FTO. We've flipped 5G. 17)  Please check the reference list; ex; ref 5, 17, 20, 23, and others We have corrected it 18)  The English should be revised by a native English speaker We have corrected it

Reviewer 2 Report

The topic of the manuscript entitled “FTO regulated intramuscular fat by targeting APMAP gene via an m6A-YTHDF2–dependent manner in rex rabbits” is interesting and meaningful. This manuscript mainly describes the regulation of m6A on adipocyte differentiation and its effect on intramuscular fat deposition through FTO-YTHDF2-APMAP signal pathway, which provides a new way for animal breeding. I recommend that this manuscript be published on cells. but some changes need to be made before publishing

Abstract: 

Analysis result showed there is a positive correlation”, The result is unclear.

Introduction:

YTHDF2 is an important factor in the manuscript, but the description of YTHDF2 in the Introduction is too little.

Material and methods

Line 49 “provided novel insight” a should be added before novel, or change insight to insights.

Line 102- There should be a space in front of (IMF).

The content of Western blotting is too little. It should be more detailed

Results

Line 125 -There should be a space in front of C.

Line 197-One more space in front of (G, H, I)

Line 198- P≤0.05 should be delated, Because the result of P≤0.05 does not appear in Figure 6.

Discussion

Line 232- The writing of m6A is inconsistent.

Line 257- “So, high content of APMAP means high intramuscular fat content.” This statement is incorrect. According to the context, high expression of APMAP gene means high intramuscular fat content.

Author Response

Thank you very much for your guidance to my manuscript. We have made relevant revisions according to the proposed guidance. I hope to be approved by editor and reviewer.

The specific reply is as follows:

The topic of the manuscript entitled “FTO regulated intramuscular fat by targeting APMAP gene via an m6A-YTHDF2–dependent manner in rex rabbits” is interesting and meaningful. This manuscript mainly describes the regulation of m6A on adipocyte differentiation and its effect on intramuscular fat deposition through FTO-YTHDF2-APMAP signal pathway, which provides a new way for animal breeding. I recommend that this manuscript be published on cells. but some changes need to be made before publishing

Abstract: 

“Analysis result showed there is a positive correlation”, The “result” is unclear.

We have corrected it

Introduction:

YTHDF2 is an important factor in the manuscript, but the description of YTHDF2 in the Introduction is too little.

We have corrected it

Material and methods

Line 49 “provided novel insight” a should be added before novel, or change insight to insights.

We have corrected it

Line 102- There should be a space in front of (IMF).

We have corrected it

The content of Western blotting is too little. It should be more detailed

We have corrected it

Results

Line 125 -There should be a space in front of C.

We have corrected it

Line 197-One more space in front of (G, H, I)

We have corrected it

Line 198- P≤0.05 should be delated, Because the result of P≤0.05 does not appear in Figure 6.

We have corrected it

Discussion

Line 232- The writing of m6A is inconsistent.

We have corrected it

Line 257- “So, high content of APMAP means high intramuscular fat content.” This statement is incorrect. According to the context, high expression of APMAP gene means high intramuscular fat content.

We have corrected it
